# Adaptation Strategies to High Hydrostatic Pressures in *Pseudothermotoga* species Revealed by Transcriptional Analyses

**DOI:** 10.3390/microorganisms11030773

**Published:** 2023-03-17

**Authors:** Romain Fenouil, Nathalie Pradel, Hassiba Belahbib, Marie Roumagnac, Manon Bartoli, Wajdi Ben Hania, Yann Denis, Marc Garel, Christian Tamburini, Bernard Ollivier, Zarath Summers, Fabrice Armougom, Alain Dolla

**Affiliations:** 1Aix Marseille Univ., Université de Toulon, CNRS, IRD, MIO, Marseille, France; 2Institut de Microbiologie de la Méditerranée, CNRS—Aix Marseille Université, Marseille, France; 3LanzaTech, Illinois Science and Technology Park, Skokie, IL 60077, USA

**Keywords:** hydrostatic pressure, adaptation, *Pseudothermotoga* sp.

## Abstract

*Pseudothermotoga elfii* strain DSM9442 and *P. elfii* subsp. *lettingae* strain DSM14385 are hyperthermophilic bacteria. *P. elfii* DSM9442 is a piezophile and was isolated from a depth of over 1600 m in an oil-producing well in Africa. *P. elfii* subsp. *lettingae* is piezotolerant and was isolated from a thermophilic bioreactor fed with methanol as the sole carbon and energy source. In this study, we analyzed both strains at the genomic and transcriptomic levels, paying particular attention to changes in response to pressure increases. Transcriptomic analyses revealed common traits of adaptation to increasing hydrostatic pressure in both strains, namely, variations in transport membrane or carbohydrate metabolism, as well as species-specific adaptations such as variations in amino acid metabolism and transport for the deep *P. elfii* DSM9442 strain. Notably, this work highlights the central role played by the amino acid aspartate as a key intermediate of the pressure adaptation mechanisms in the deep strain *P. elfii* DSM9442. Our comparative genomic and transcriptomic analysis revealed a gene cluster involved in lipid metabolism that is specific to the deep strain and that was differentially expressed at high hydrostatic pressures and might, thus, be a good candidate for a piezophilic gene marker in *Pseudothermotogales*.

## 1. Introduction

Several studies have highlighted the fact that hydrostatic pressure affects the activity of numerous key processes and cellular components of deep-living microorganisms [1,2,3]. Depending on the genera investigated, piezophilic adaptation requires modifications of the genome, fine tuning of the gene expression, modifications of the structure of key proteins, or stress-like physiological cell responses [4]. An important finding is that osmolyte-type co-solutes, which are termed “piezolytes”, accumulate in the cells and counteract the pressure effects [5]. Increasing hydrostatic pressure has been observed to lead to intracellular accumulations of β-hydroxybutyrate, ectoine, and mannosyl-glycerate in *Photobacterium profundum*, *Alcanivorax borkumensis*, and the hyperthermophilic piezophilic archaeon *Thermococcus barophilus*, respectively [6,7,8]. Glutamate has also been suggested to be a piezolyte in the meso- and piezophilic *Desulfovibrio hydrothermalis* and *D. piezophilus* species [9,10]. Recently, it has been suggested that glutamate optimizes the enzymatic activities under high hydrostatic pressure [11]. Furthermore, it has been shown that several microorganisms adapt to high pressure by changing their energy metabolism; for instance, in the anaerobic sulfate-reducing bacterium *D. piezophilus*, transcriptomic and biochemical analyses have shown that the metabolite cycling H_2_/Formate is required for energy-conservation at high hydrostatic pressure [10,12]. Transcriptome analyses of the piezo-hyperthermophile *Thermococcus piezophilus* and *Pyrococcus yayanosii* highlighted the crucial role played by energy-conservation processes in the adaptative response of these archaea to hydrostatic pressure [13,14]. In addition, several studies have reported an increase in the biosynthesis of unsaturated and branched-chain fatty acids at high hydrostatic pressure for several piezophile and piezotolerant microorganisms, indicating that maintaining the fluidity of the membrane is an essential strategy for microorganisms to cope with high hydrostatic pressure [15,16,17].

Microbes inhabiting deep-oil reservoirs must be able to cope with high in situ temperatures and pressures. Considering that the effects of high pressure on these microbes are rarely studied, any new research into this area is likely to lead to the discovery of unique adaptation strategies to these extreme conditions [18]. Members of the order *Thermotogales*, including the genus *Pseudothermotoga*, are ubiquitous and predominant in global oil reservoirs [19]. The order *Thermotogales* belongs to the phylum Thermotogae, which comprises mesophilic, thermophilic, and hyperthermophilic bacteria, all of which are characterized by a toga-like sheath surrounding the cell. This phylum has been redefined, based on a genomic analysis of its members, in four separate orders, including *Thermotogales*, and five families including *Thermotogaceae*, which contains two genera, *Thermotoga* and *Pseudothermotoga* [20]. The *Pseudothermotoga* genus is composed of rod-shaped anaerobic bacteria with an optimal growth temperature in the range of 65–70 °C. To date, the DNA genome sequences of eight *Pseudothermotoga* strains have been determined (https://www.ncbi.nlm.nih.gov/ (accessed on 10 March 2023)) [21]. A study by Roumagnac and collaborators [22] compared the growth and cell phenotypes of two strains of *Pseudothermotoga elfii* species for different hydrostatic pressures, namely, *P. elfii* type strain DSM9442 isolated from a deep, oil-producing well (1600–1900 m depth) in Africa with an in situ temperature of 68 °C [23] and *P. elfii* subsp. *lettingae* DSM14385 isolated from a thermophilic sulfate-reducing bioreactor operated at 65 °C and at atmospheric pressure with methanol as the sole substrate [24]. Their results showed that the physiology of cells varied differently in response to variations in hydrostatic pressure. They found evidence for the piezophilic nature of *P. elfii* DSM9442, with an optimal hydrostatic pressure for growth of 20 MPa and an upper limit of 40 MPa, while *P. elfii* subsp. *lettingae* was found to be piezotolerant, with growth occurring up to 20 MPa, and with no growth at 30 MPa. Metabolite analyses revealed that the metabolism of *P. elfii* DSM9442 was optimal at 20 MPa and that the propionate production was greater at high hydrostatic pressure than at atmospheric pressure. Slight modifications of the membrane composition occurred in both strains in response to changes in hydrostatic pressure [22]. In *P. elfii* DSM9442, the formation and viability of chained cells increased with increasing hydrostatic pressure, indicating that chain formation might be a protective mechanism [22]. These strains are closely phylogenetically related and provide us with new insights into how these microorganisms adapt to high hydrostatic pressures. Here, we compared the genomes of the same two strains and analyzed their response to changes in hydrostatic pressure by comparing the transcriptomes of the deep strain *P. elfii* DSM9442 grown at 0.1 MPa (atmospheric pressure) vs. 20 MPa (optimal growth pressure), 30 MPa, and 40 MPa (maximum growth pressure) and the transcriptomes of surface strain *P. elfii* subsp. *lettingae* grown at 0.1 MPa and 20 MPa. The results suggest common traits of adaptation to increasing hydrostatic pressure in both strains, as well as species-specific adaptations, and highlight the pivotal role played by aspartate in the adaptation of the deep strain *P. elfii* DSM9442 to high hydrostatic pressure.

## 2. Materials and Methods

### 2.1. Comparative Genomic Analysis

The genomes of cultivated strains *T. elfii* DSM9442 and *T. elfii* subsp. *lettingae* DSM14385 were retrieved from the NCBI FTP site (ftp://ftp.ncbi.nlm.nih.gov/) under the accession numbers AP014507 and CP000812, respectively. The gene/protein predictions and genome annotation were performed by using the Prokka package [25]. The clustering of orthologous proteins for core/pan-genomic analysis was carried out by using the Roary pan genome pipeline v 3.6.1 [26] and applying a 50% minimum of sequence identity and coverage, as recommended by Tettelin and colleagues [27]. The Roary pipeline generated a core gene alignment and gene presence/absence matrix that were used by Roary scripts to retrieve strain-specific genes (not belonging to any orthologous cluster) of *T. elfii* DSM9442 and *T. elfii* subsp. *lettingae* DSM14385. Each strain-specific protein was carefully checked for functional annotation by using the Conserved Domains Database at NCBI (ftp://ftp.ncbi.nlm.nih.gov/pub/mmdb/cdd (accessed on 6 January 2020)), which includes the COG (Clusters of Orthologous Groups of proteins) functional database (rpsblast of the predicted proteins against COG profiles included in the COG annotation selected for E-value below 10^−5^). The collinearity of genomes was evaluated via MAUVE tool alignment [28].

### 2.2. Growth under High Hydrostatic Pressure

*P. elfii* DSM9442 and *P. elfii* subsp. *lettingae* (DSM14385) were grown anaerobically at 65 °C in Hungate tubes (18 mL) or rubber-stoppered Schott bottles (100 mL) fully filled with the medium previously described [22], which were inoculated at a 1:10 ratio. Tubes or bottles were inserted into 500 mL high-pressure bottles (HPBs) (HPB-500; Top Industrie, Vaux-le-Pénil, France), and the hydrostatic pressure was controlled by using a piloted pressure generator (PMHP 600-600; Top Industrie, Vaux-le-Pénil, France) connected to the HPBs [29]. The hydrostatic pressure applied to the HPBs was transmitted to the culture via the septum of the Hungate tubes or via the Schott bottle screw cap (Duran group, Germany), which was modified to include a rubber septum (Fisher Scientific, France). HPBs were incubated at 65 °C. Two consecutive subcultures were grown from frozen stocks in fully filled 18 mL Hungate tubes for 60 h at 65 °C at hydrostatic pressures of 0.1, 20, 30, and 40 MPa for *P. elfii* DSM9442 and of 0.1 and 20 MPa for *P. elfii* subsp. *lettingae*. Then, each culture was used to inoculate 100 mL of fresh medium in Schott bottles for growth under the same temperature and hydrostatic pressures. Cells were harvested in the mid-exponential growth phase and used for total RNA preparation. Five replicates of each culture were performed.

### 2.3. Aspartate Measurements

Intracellular aspartate quantitation was performed at 0.1 and 30 MPa on cells in mid-exponential growth cultured in 100 mL bottles at the indicated hydrostatic pressures (by using the same protocol as above for the RNA preparation). Triplicate cultures were performed. Cells were pelleted via centrifugation at 9000× *g* for 15 min at 4 °C and then washed once with 1 mL of 200 mM NaCl. After centrifugation, the pellet was suspended with 0.5 mL hot distilled water and incubated for 15 min at 100 °C, followed by incubation on ice for 10 min. Then, the extract was centrifuged at 15,000× *g* for 5 min at 4 °C to eliminate cell debris, and the supernatant was aliquoted, snap-frozen in liquid nitrogen, and stored at −80 °C. Aspartate was measured by using the fluorometric Aspartate assay Kit (Abcam, Cambridge, UK). Samples were plated in 96-well black microplates (Costar^TM^ 96-well Assay Plates, Black Polystyrene) for fluorescence readings at Ex/Em 535 nm/587 nm on a TECAN-Spark Infinite M200 plate reader. In the meantime, the number of cells on each sample was counted as previously described [22] in order to relate the aspartate concentration to the cells amount. Experiments were performed in triplicate.

### 2.4. RNA-Seq Experiments

For RNA preparation, cells were grown in 120 mL flasks at the desired hydrostatic pressure until reaching mid-exponential growth. Then, cells were harvested via centrifugation at 6800× *g* for 15 min at 4 °C; the pellet was resuspended in 400 µL of 10 mM Tris-HCl (pH 8.0) and spun for 2 min at 12,000× *g*. The supernatant was removed, and the pellet was frozen in liquid nitrogen before being placed in −80 °C storage until use. RNA samples were prepared by using the Maxwell 16 LEV samples kit (Promega), and the quality was checked by using the Experion RNA analysis kit (Bio-Rad).

RNA library preparations and sequencing reactions were conducted at GENEWIZ, LLC. (South Plainfield, NJ, USA). Ribosomal RNA samples were depleted by using the Illumina Ribo-zero Plus rRNA depletion kit (Illumina, San Diego, CA, USA), and the sequencing libraries were prepared by using the NEBNext Ultra RNA Library Prep Kit for Illumina by following the manufacturer’s recommendations (NEB, Ipswich, MA, USA). Sequencing libraries were validated by using a DNA Chip on the Agilent 2100 Bioanalyzer (Agilent Technologies, Palo Alto, CA, USA) and were quantified by using a Qubit 2.0 Fluorometer (Invitrogen, Carlsbad, CA, USA), as well as via quantitative PCR (Applied Biosystems, Carlsbad, CA, USA). The sequencing libraries were multiplexed and clustered on six lanes of a flowcell. After clustering, the flowcell was loaded on the Illumina HiSeq 2500 instrument according to the manufacturer’s instructions. The multiplexed libraries were sequenced by using paired-end 150 cycle chemistry for the HiSeq 2500 (Illumina). Image analysis and base calling were conducted via the HiSeq Control Software (HCS) v2.2.68 on the HiSeq 2500 instrument. Raw sequence data generated from Illumina HiSeq 2500 was converted into fastq files and de-multiplexed by using Illumina CASAVA 1.8.2 software. One mismatch was allowed for index sequence identification. RNA-seq data were submitted to the NCBI Gene Expression Omnibus (GEO, https://www.ncbi.nlm.nih.gov/geo/ (accessed on 25 February 2023)) under the accession number GSE226101.

### 2.5. Bioinformatics Analysis of RNA-Seq Data (Integrated Workflow for RNA-Seq Processing)

A Nextflow pipeline was developed to process sequenced samples [30]. By integrating standard RNA-seq data processing tools, this pipeline performs various QC assessments of the sequencing data, aligns reads to the reference genome, and provides a read-count-based comparison of samples (correlation matrix and multidimensional scaling). During pre-processing, the base-quality of raw and trimmed fastq files was analyzed with “fastqc 0.11.5”. Sequences were trimmed with “TrimGalore 0.4.5” to remove low-quality bases (phred score < 20) and eventual sequencing adapters before alignment to the corresponding reference genome (*P. elfii*: GCF_000504085.1_ASM50408v1, *P. lettingae*: GCF_000017865.1_ASM1786v1) with “STAR 2.5.3a” (“alignMatesGapMax” = 10,000, and disabling splicing alignment by setting “alignSJoverhangMin” = 999 and “alignIntronMax” = 1) [31]. Post-alignment statistics were computed with the “RSeQC 2.6.4” [32], “preseq 2.0” [33], and “R 3.4.4” package “dupRadar” [34]. A summary of read counts for annotation categories was computed with “featureCounts” [35] from “subread 1.6.0”. Finally, a FPKM score was computed with “StringTie 1.3.3b” [36] to allow the computation of MDS plots and correlation maps between replicates. All results were compiled into an HTML report by using “MultiQC 1.6” [37].

During the analysis, the selected replicates were submitted to a custom R script for differential expression analysis by using the “edgeR” package [38]. Two sets of samples, including all RNA-seq replicates for *P. elfii* DSM9442 and *P. elfii* subsp. *lettingae* were processed, revealing a very high overall quality of samples and an extensive coverage of reference genomes. Two samples from each set showed deviations from their corresponding replicates and were discarded from further analyses. All conditions retained at least 3 replicates for downstream analyses. Genes were considered significantly differentially expressed if their log_2_-fold change was above 1.3 and the Benjamini–Hochberg corrected *p*-value was below 0.01.

To provide functional information about differentially expressed genes (DEG), the full KEGG (Kyoto Encyclopedia of Genes and Genomes) database was downloaded [39,40,41]. Sequences for all annotations in reference genomes were aligned to KEGG Orthology (KO) annotation sequences by using “blastx” (E-value < 1 × 10^−10^, BitScore > 50) [42]. Resulting KO molecular function annotations were added to the original annotation files and reported as part of the results of differential expression analyses.

## 3. Results and Discussion

### 3.1. Comparative Genomics of the Deep and Surface *P. elfii* Strains

We conducted a comparative genomics study between the piezophilic strain *P. elfii* DSM9442 (accession number: AP014507) (optimal growth at 20 MPa) and the piezotolerant strain *P. elfii* subsp. *lettingae* DSM14385 (accession number: CP000812) (optimal growth at 0.1 MPa). The genetic closeness of the two genomes in combination with different sensitivities of the strains to high hydrostatic pressure provided a valuable framework to identify potential genetic determinants associated with the piezophilic lifestyle in the genus *Pseudothermoga*.

As previously reported, *P. elfii* and *P. elfii* subsp. *lettingae* have intra-species relationships with an average nucleotide identity score of 99.12% [21]. We found a strong genome collinearity (results not shown), indicating a high degree of gene order conservation. The difference in genome size between *P. elfii* DSM9442 and *P. elfii* subsp. *lettingae* DSM14385 is almost negligible, with 2,169,860 and 2,135,342 base pairs, respectively, while the guanine–cytosine (GC) content is the same (39%). Consequently, the number of coding sequences (CDS) is very similar between the two strains, with 2063 and 2047 CDS for *P. elfii* and *P. elfii* subsp. *lettingae*, respectively.

A comparison of all CDS (i.e., the predicted proteome) between both species resulted in the identification of orthologous proteins shared by both strains (core-genome) and of those unique to each strain (strain-specific). Our main focus was the identification of the strain-specific genes/proteins in the piezophilic strain *P. elfii* DSM9442 and, therefore, lost (those originating from the ancestral genome) in the surface strain *P. elfii* subsp. *lettingae*. These potential piezophilic markers will be the particular focus of the transcriptomic analysis (see below).

We identified a total of 77 genes that were specific to *P. elfii* DSM9442 (absent in *P. elfii* subsp. *lettingae*). They were mostly gathered in clusters and operons that were related mainly to carbohydrate transport and metabolism, cell envelope biogenesis, and amino acid transport and metabolism (Appendix A). In total, 16 strain-specific genes were annotated as hypothetical proteins. Several clusters contained genes encoding ABC transporters, which are known to be involved in nutrient uptake but could also be linked to environmental stress responses [43]. Clusters 1 to 4 provided different ABC transporters for the transport and metabolism of sugar (Xylulose, L-Arabinose, or D-ribose due to strong operon homology) and phosphate (under phosphate starvation via the uptake of Glycerol-2P and Glycerol 3-P), for dipeptide transport (Cluster 3) and the transport of branched-chain amino acids [44] including L-Leucine, L-Phenylalanine, L-Isoleucine, and L-Valine (Cluster 4, LivKHMGF operon). Interestingly, the LivKHMGF operon was present in multiple copies in both strains, although Cluster 4, which was present solely in the deep strain *P. elfii* DSM9442, bore little similarity with the other Liv operons (under the homology cutoff). This Liv operon could be a paralog (Liv duplicate) under a neo-functionalization or pseudogenization process, which would explain the sequence divergence with respect to the other Liv copies. We identified another transport system in Cluster 7 that exhibited a TRAP system permease (DctPQM). Previous studies have shown that this system has a high affinity for importing C4-dicarboxylates, such as malate succinate and fumarate [45].

Interestingly, the genes involved mainly in fatty acid biosynthesis (AcP, FaD, and FabG; Cluster 6) were found to be specific to the deep strain, *P. elfii*, with the closest related proteins (by using a similarity-based search) belonging to *Firmicutes* representatives and to the anaerobic thermophilic bacterium *Caldithrix abyssi*, which was isolated from a depth of 3000 m [46].

Defense mechanisms were represented in Cluster 8 by the CRISPR-associated RAMP cmr system, which protects bacteria against foreign RNA from plasmids or viruses. It has also been shown to possess the ability to cleave endogenous RNA of *Pyrococcus furiosus* in vivo [47]. We also found a specific McrA endonuclease acting against foreign DNA. In the identified clusters, we found several glycosyl transferases for the initiation and elongation of glycan chains, flippase for the translocation of lipid-linked oligosaccharides [48], and a transcriptional regulator mainly for drug resistance.

We identified a total of 43 genes that were specific to *P. elfii* subsp. *lettingae*, 19 of which were found in 4 clusters that are related mainly to cell envelope biogenesis, amino acid transport (Cluster 3), and carbohydrate metabolism (Appendix A). Cluster 4 is composed of genes involved in poly-γ-glutamate (PGA) synthesis. While PGA can bind to peptidoglycan to facilitate nutrient uptake, it can also be released into the environment to sequester toxic metal ions, decrease salt concentration, and protect against adverse conditions [49].

### 3.2. Overview of the Modifications Induced by Hydrostatic Pressure at the Transcriptome Level of *P. elfii* DSM9442 and *P. elfii* subsp. *lettingae*

Adaptation to high hydrostatic pressure should involve transcriptional regulation processes. In order to elucidate the hydrostatic pressure adaptation strategies of *P. elfii* DSM9442 and *P. elfii* subsp. *lettingae*, the transcriptome of cells in mid-exponential growth was analyzed via RNA-seq. The DEG values of *P. elfii* DSM9442 (grown at 0.1, 20, 30, and 40 MPa) and those of *P. elfii* subsp. *lettingae* (grown at 0.1 and 20 MPa) are listed in Appendix A, respectively. By using 0.1 MPa as the reference for the deep strain *P. elfii* DSM9442, a total of 58 genes was differentially expressed at 20 MPa (optimal hydrostatic pressure for growth), 211 genes were differentially expressed at 30 MPa, and 124 genes were differentially expressed at 40 MPa (maximum hydrostatic pressure for growth) (Figure 1).

It should be noted that a cluster of three genes (locus tags TEL01S_RS06040–TEL01S_RS06050) which encode hypothetical proteins exhibited a more complex expression pattern in that they were under-expressed at 20 and 40 MPa vs. 0.1 MPa and over-expressed at 30 MPa vs. 0.1 MPa. A similar number of genes were over-/under-expressed at each pressure, namely 28/30 genes at 20 MPa, 103/108 at 30 MPa, and 53/71 at 40 MPa (Figure 1). The number of DEG was higher at 30 and 40 MPa compared to 20 MPa, which shows that the response of the cells to hydrostatic pressures exceeding the optimal pressure for growth requires a larger variation in the transcriptome. It can be explained by the phenotypic response previously described, showing that at 30 and 40 MPa, the final biomass and growth rates were lower than at 20 MPa and 0.1 MPa [22].

In the surface strain, *P. elfii* subsp. *lettingae*, 31 genes were differentially expressed between 0.1 MPa and 20 MPa, with 21 being over- and 10 being under-expressed at 20 MPa (Appendix A). This lower number of DEG (compared to *P. elfii* DSM9442) in response to pressure could be due to the more similar growth characteristics of *P. elfii* subsp. *lettingae* at 0.1 MPa and 20 MPa [22]

### 3.3. Differentially Expressed Genes in *P. elfii* DSM9442 and *P. elfii* subsp. *lettingae* in Response to Hydrostatic Pressure

We conducted a systematic comparison between detected transcripts of *P. elfii* DSM9442 that were cultured at 0.1 MPa (atmospheric pressure) versus detected transcripts when the strain was grown at 20 MPa (optimal pressure for growth), 30 MPa, and 40 MPa (maximum pressure for growth). Overall, the expression patterns for *P. elfii* DSM9442 showed few overlaps for the different pressures (Figure 1). Only 13 genes were found to be similarly differentially expressed for all 3 pressures (Appendix A, Figure 1), with 6 over- and 7 under-expressed. This included two genes involved in amino acid metabolism, namely, TEL01S_RS09135, which encodes a protein belonging to the alanine-glyoxylate aminotransferase family, and TEL01S_RS06965, which encodes an aspartate kinase. None of the orthologs of these genes was found to be differentially expressed in the surface strain *P. elfii* subsp. *lettingae*, suggesting that this response is specific to the deep strain.

When the transcriptomes of cells grown at 0.1 MPa and 20 MPa are compared, 21 genes were specifically over- and 16 were under-expressed at 20 MPa (Figure 1). Among the over-expressed genes, 14 (locus tags TEL01S_RS00165 to TEL01S_RS00175, TEL01S_RS00525 to TEL01S_RS00535, and TEL01S_RS08670 to TEL01S_RS08705) encode proteins of sugar transport and metabolism, while two genes (locus tags TEL01S_RS06345 and TEL01S_RS06350) encode an ABC-2 type transporter system. Furthermore, five of the under-expressed genes encode proteins linked to sugar transport and metabolism, and a cluster of five genes (locus tags TEL01S_RS05715 to TEL01S_RS05735) encode proteins involved in a branched-chain amino acid transport system (Appendix A).

At even higher pressures of 30 and 40 MPa, we found a total of 67 genes similarly differentially expressed that were not differentially expressed at 20 MPa (Figure 1), 29 of which were over-expressed and involved mainly in the transport system and amino-acid metabolism. They include genes with locus tags TEL01S_RS00940 to TEL01S_RS00950, which encode a peptide/nickel transport system; TEL01S_RS01770 to TEL01S_RS01785, which are genes linked to the arginine biosynthesis pathway; TEL01S_RS03985 to TEL01S_RS03995, which encode a branched-chain amino acid transport system; TEL01S_RS03635, which encodes an aspartate kinase LysC; and TEL01S_RS06945, which encodes the aspartate-semialdehyde dehydrogenase Asd linked to the glycine, serine, and threonine metabolism. Genes involved in amino acid metabolism and transport systems were also identified among the 38 genes under-expressed at 30 and 40 MPa (Appendix A). These results support our hypothesis that the transport systems and amino acid metabolism play a pivotal role in the adaptation mechanisms of *P. elfii* DSM9442 to high hydrostatic pressures.

In total, 131 genes are specifically, differentially expressed between 0.1 and 30 MPa. Among those what were over-expressed, 9 encode proteins of the sugar transport systems; 14 encode other ABC-type transport systems, with those with locus tags TEL01S_RS02380 to TEL01S_RS02390 encoding a putative polar amino acid transport system; and 6 genes are involved in amino acid metabolism (Appendix A). Interestingly, genes TEL01S_RS06225 to TEL01S_RS06235—which encode proteins AcpP, FadD, and FabG, which are involved in fatty acid (FA) biosynthesis and are absent in *P. elfii* subsp. *lettingae* genome—and TEL01S_RS09220—which encodes the UDP-N-acetylmuramoyl-tripeptide--D-alanyl-D-alanine ligase MurF, which is involved in peptidoglycan biosynthesis—were under-expressed.

A total of 36 genes was specifically differentially expressed between 0.1 and 40 MPa. A *fabG* homolog (TEL01S_RS00260) was over-expressed, while the cluster TEL01S_RS08165 to TEL01S_RS008175, which is absent in the *P. elfii* subsp. *lettingae* genome and which encodes an ABC-type peptide transport system, and genes TEL01S_RS09090–TEL01S_RS09095, which encode a 2-oxoacid:ferredoxin oxidoreductase linked to the TCA cycle and to gluconeogenesis, were under-expressed. Hence, those genes involved in FA and peptidoglycan synthesis, in the transport of peptide, and in amino acid metabolism tend to respond to the highest pressures.

In *P. elfii* subsp. *lettingae*, a total of 31 genes was found to be differentially expressed at 20 MPa (maximum pressure for growth), 21 of which were over-expressed, while 10 were under-expressed (Appendix A). Of the former, nine genes of the cluster TEL01S_RS00925–TEL01S_RS00980 encode proteins involved in ABC membrane transport and the metabolism of sugars (pentose–phosphate pathway), gene TEL01S_RS01250 encodes a protein linked to peptidoglycan biosynthesis, and gene TEL01S_RS08015 encodes a short-chain FA transporter. Of the 10 under-expressed genes, TEL01S_RS01020–TEL01S_RS01025 encode a peptide/nickel transport system.

Overall, these results show that genes that are involved in membrane transport play pivotal roles in the response of both *P. elfii* strains to high hydrostatic pressures. Notably, these gene expression patterns highlight the reshuffle of the branched-chain amino acid transport systems in the deep strain *P. elfii* DSM9442 in response to pressure. Moreover, the response in this strain also involved numerous genes linked to amino acid metabolism. In contrast, few of the differentially expressed genes in the surface strain *P. elfii* subsp. *lettingae* were linked to amino acid metabolism. Of the 31 genes differentially expressed in *P. elfii* subsp. *lettingae*, 11 orthologs were found to be differentially expressed in *P. elfii* DSM94442 as well (Appendix A), although 9 of them were found to be expressed in opposing directions, which suggests a different pressure response altogether in these two strains.

### 3.4. DEG-Related KEGG Pathway Distribution in *P. elfii* DSM9442 and *P. elfii* subsp. *lettingae*

To obtain a more integrated view of the transcriptomic variations in response to hydrostatic pressure, we analyzed the distribution of KEGG pathways that are related to the list of differentially expressed genes. A KEGG pathway was considered significantly enriched if its percentage was higher among the DEG-related vs. genome-related KEGG-pathways (one-tail z-test, *p*-value < 0.05). By examining the patterns of DEG-related KEGG-pathway enrichments for *P. elfii* DSM9442 as a function of hydrostatic pressure, we found that the pathways involved in amino acid metabolism, the biosynthesis of other secondary metabolites, the metabolism of terpenoid and polyketides, and the metabolism of cofactors and vitamins were among the most enriched pathways, irrespective of the hydrostatic pressure (Figure 2). At 20 MPa, DEG-related KEGG pathways involved in membrane transport and amino acid metabolism were the most enriched (Figure 2A). At 30 and 40 MPa, the total number of enriched DEG-related KEGG-pathways was more than twice the number at 20 MPa (Figure 2B,C vs. Figure 2A). A similar KEGG-pathway enrichment was highlighted between 30 and 40 MPa. In addition, the KEGG pathways involved in lipid metabolism were enriched only for hydrostatic pressures ≥30 MPa.

In *P. elfii* subsp. *Lettingae*, the enrichment pattern of DEG-related KEGG pathways revealed the membrane transport pathway as the most enriched pathway as for for *P. elfii* DSM9442 at 20 MPa (Figure 3).

The pathway involved in folding, sorting, and degradation was among the most enriched in *P. elfii* subsp. *lettingae* but not in *P. elfii* DSM9442. The latter could be linked to the piezotolerant capability of *P. elfii* subsp. *lettingae* [22], which allows bacteria to fold and/or degrade macromolecules at high hydrostatic pressures. Interestingly, DEG-related KEGG pathways involved in amino acid metabolism were among the most enriched pathways in *P. elfii* DSM9442 but not in *P. elfii* subsp. *lettingae*.

Overall, this analysis reveals the common traits of adaptation to increasing hydrostatic pressure in both strains, namely, variations in transport membrane or carbohydrate metabolism. There were also several species-specific adaptations: variations in amino acid metabolism, lipid metabolism, or biosynthesis of secondary metabolites were mostly found in the deep strain *P. elfii* DSM9442, while changes to the folding, sorting, and degradation pathways were found in the surface strain *P. elfii* subsp. *lettingae*. This suggests that members of the genus *Pseudothermotoga* activate both similar and specific global functions in response to the hydrostatic pressure increases.

### 3.5. Variations in the Amino Acid Metabolism in *P. elfii* in Response to Pressure Changes

As mentioned above, the deep strain *P. elfii* DSM9442 responds to increases in hydrostatic pressure through changes in the metabolic pathways involved in amino acid metabolism. Twenty-eight genes encoding proteins that are involved in amino acid metabolism were detected as differentially expressed depending on the hydrostatic pressure (Table 1 and Appendix A), corresponding mostly to pathways linked to arginine biosynthesis, glycine serine threonine metabolism, cysteine methionine metabolism, and lysine biosynthesis.

Notably, the *lysC* gene TEL01S_RS03635 coding for an aspartate kinase and the *asd* gene TEL01S_RS06945 encoding an aspartate-semialdehyde dehydrogenase, were over-expressed at both 30 and 40 MPa, while the *thrA* gene TEL01S_RS06965 encoding a bifunctional aspartokinase/homoserine dehydrogenase was over-expressed at 20, 30, and 40 MPa (Table 1). These gene products are involved in the synthesis of aspartate-4-semialdehyde from aspartate (Figure 4).

Aspartate-4-semialdehyde is a metabolite that is involved in the biosynthesis of methionine, threonine, and lysine. In addition, genes *dapA*, *B*, and *D* (TEL01S_RS00130, TEL01S_RS03640, and TEL01S_RS10080), which are involved in lysine biosynthesis, were over-expressed at 30 MPa (Figure 4, Table 1), while genes *metB*, *metH*, *mtnK*, and *mtnA* (TEL01S_RS05650, TEL01S_RS06970, TEL01S_RS08685, and TEL01S_RS08690), which are involved in methionine metabolism, were over-expressed at 20 and 30 MPa. The gene cluster TEL01S_RS01770–TEL01S_RS01785, which encodes ArgGHCJ, was over-expressed at 30 and 40 MPa (Table 1). The N-acetylglutamate synthase/glutamate N-acetyltransferase encoded by *argJ* and the N-acetyl-gamma-glutamyl-phosphate reductase encoded by *argC* are involved in the synthesis of ornithine, a component of the urea cycle from glutamate. The arginosuccinate synthase and the arginosuccinate lyase encoded, respectively, by *argG* and *argH* are enzymes of the urea cycle, allowing arginine synthesis from aspartate (Figure 4). These variations in gene expression suggest that the arginine biosynthesis pathway from aspartate was boosted at high pressures. However, as the urea cycle produces fumarate, we could also hypothesize that feeding of the citrate cycle via fumarate is boosted when cells are grown at elevated hydrostatic pressure. This would not only provide energy to the cells but also contribute to gluconeogenesis via the production of oxaloacetate, as aspartate has been shown to be a glucogenic amino acid [50]. The gluconeogenesis pathway end-product, glucose-6-phosphate, is used to form six-carbon or five-carbon sugars needed for cell envelope synthesis. These include precursors of the peptidoglycan layer located in the periplasmic space, the lipopolysaccharides present in the outer membrane, and several sugars located in other cell compartments [51,52]. In a previous report, Roumagnac and collaborators [22] observed that under increasing hydrostatic pressure, *P. elfii* strains formed chains containing up to 15 cells per chain, possibly as a protective mechanism. An increased gluconeogenesis from aspartate at high pressures would, thus, allow cells to synthesize those molecules required for the cell envelope to facilitate chain formation.

Overall, the biosynthesis pathways of lysine, arginine, methionine, threonine, glycine, and serine from aspartate appeared to be boosted when cells were grown at high hydrostatic pressures, which is indicative of the pivotal role played by aspartate in the pressure response of the deep strain *P. elfii* DSM9442. In addition, the cluster TEL01S_RS02380–TEL01S_RS02390, which encodes a polar amino acid transport system, was over-expressed at 30 MPa vs. 0.1 MPa, which may suggest that this transport system is activated at 30 MPa to optimize the import of aspartate into the cells. To determine whether the intracellular abundance of aspartate varied with hydrostatic pressure, we quantified the intracellular aspartate concentration in *P. elfii* DSM9442 at 0.1 and 30 MPa, i.e., the pressure conditions for which major differences in gene expression were found. At 30 MPa, intracellular aspartate in *P. elfii* DSM9442 was 4 times more abundant compared to 0.1 MPa (3.5 vs. 14 nmol/10^6^ cells) (Figure 5). The transcriptomic data provided no evidence for any increase in the aspartate biosynthesis pathway. Therefore, the higher aspartate abundance in cells grown at elevated hydrostatic pressure may be attributed to a more efficient aspartate import. These data support our hypothesis that aspartate is indeed a key metabolite for several pathways and adaptation to high pressures in *P. elfii* DSM94442.

In the surface strain *P. elfii* subsp. *lettingae*, only one gene of the amino acid metabolism, which is involved in glycine biosynthesis (gene *soxA*, locus tag TEL01S_RS05600), was found to be over-expressed at 20 MPa vs. 0.1 MPa, while the orthologous gene in *P. elfii* DSM9442 (TEL01S_RS05590) was also over-expressed at both 30 and 40 MPa. Hence, in contrast to *P. elfii* DSM9442, amino acid metabolism in *P. elfii* subsp. *lettingae* does not seem to play an important role in the response of the strain to high hydrostatic pressures.

Under these experimental conditions, peptides and/or amino acids contained in biotrypcase are the only sources of carbon and energy available to the cells [22]. Fermentation of these compounds leads to the production of acetate and propionate as major products and, in lower quantities, formate, succinate, isobutyrate, and isovalerate. In *P. elfii* DSM9442, the production of these fermentation end-products is unaffected by pressure, except for propionate, for which production increases slightly with increasing hydrostatic pressure [22]. It is known that the catabolic pathways of some amino acids (valine, isoleucine, threonine, and methionine) can lead to the production of propionate and ATP via propionyl-CoA [53]. As mentioned above, the threonine and methionine biosynthesis pathways from aspartate were boosted at high hydrostatic pressure. Given that valine and isoleucine are branched-chain amino acids, it is worth pointing out that the genes TEL01S_RS03990–TEL01S_RS03995, which code for a branched-chain amino acid transport system, were over-expressed at 30 MPa and 40 MPa vs. 0.1 MPa. Similarly, genes TEL01S_RS06345–TEL01S_RS06350, which encode another branched-chain amino acid transport system, were over-expressed at 20 MPa and 30 MPa. Over-expression of these genes may lead to a higher rate of import of branched-chain amino acids into the cells at high hydrostatic pressures, and through their catabolism, this could, in turn, induce a higher production of propionate in the deep strain *P. elfii* DSM9442.

### 3.6. Adaptation Involving Modifications in Membrane Composition

A cluster of three genes (TEL01S_RS06225–TEL01S_RS06235) that are specific to the deep strain *P. elfii* DSM9442 and which encode proteins AcpP, FadD, and FabG was under-expressed at 30 MPa vs. 0.1 MPa. Similarly, the gene TEL01S_RS04130, which encodes a FabG protein, was under-expressed at 30 and 40 MPa, while the paralog of *fabG* (TEL01S_RS00260) was over-expressed at 40 MPa (Appendix A). This *fabG* redundancy is likely due to gene duplication or addition events [54]. The multi-enzymes FabG, FabA, FabI, and FabB perform the elongation steps of FA synthesis [55]. FabG is a dehydrogenase that interacts with the FA chain that is bound to the acyl-carrier protein (Acp), which, in turn, binds to substrates of varying lengths. In addition, FabG often substitutes other dehydrogenases to produce secondary metabolites that may play a role in membrane permeability. It is also involved in polyhydroxyalkanoate (PHA) production, a form of carbon and energy storage under stress conditions [56]. In view of these putative functions, the differential expression of the *fabG* paralogs may explain the slight modification of the FA membrane composition in response to increasing hydrostatic pressure and/or the microscopy observations of the chain morphotype that was previously reported at the highest hydrostatic pressure [22]. It should be noted that no variation in the expression of these genes was detected in *P. elfii* subsp. *lettingae*, which is in agreement with the very low number of chains observed in the culture of this strain at high hydrostatic pressure [22]. The regulation of the *fabG* gene expression may govern modifications of the cell membrane and/or the toga, making cells more resistant to the highest hydrostatic pressures (>20 MPa). This leads us to hypothesize that each of the three FabG proteins may be structurally adapted to be active at a specific pressure (e.g., TEL01S_RS00260, which is over-represented at 40 MPa, could be adapted at high-pressure conditions). Such adaptation strategies have been previously reported, notably for the flagellar systems of the piezophilic *Photobacterium profundum* SS9 strain, in which one system was preferentially expressed at high pressure, while another was preferentially expressed at low (atmospheric) pressure [57].

## 4. Conclusions

In this paper, we reported for the first time the adaptation at the transcriptomic level to hydrostatic pressure increases of two strains from the same species, namely, the deep strain *P. elfii* DSM9442 and the surface strain *P. elfii* subsp. *lettingae*. The genetic closeness of the two strains provides a valuable framework to identify the genetic determinants that are associated with the piezophilic lifestyle in the genus *Pseudothermoga*. In that way, the gene cluster which is involved in lipid metabolism (TEL01S_RS06225–TEL01S_RS06235), which is specific to the deep strain, was differentially expressed at high hydrostatic pressures (30 and 40 MPa) and might be a good candidate for a piezophilic gene marker in the *Pseudothermotogales*. Our analysis revealed the common traits of adaptation to increasing hydrostatic pressure in both strains as well as several species-specific adaptations. Membrane transport appears to play an important role in the adaptation of both strains to 20 MPa. A key outcome of the transcriptomic analysis was that amino acid metabolism and transport appear to be crucial for the adaptation to high pressures in the deep strain *P. elfii* DSM9442. In addition, our analysis revealed that several metabolic pathways for amino acid biosynthesis that involve aspartate are boosted at high hydrostatic pressures, which suggests that a reshuffling of the amino acid metabolism may take place in response to pressure increases, which would represent an important adaptation mechanism in the deep strain *P. elfii* DSM9442. Given that the effects of high pressure on microorganisms inhabiting deep-oil reservoirs have rarely been studied, our work provides valuable new information and insights regarding the dynamics of oil reservoir microbiomes.

## Figures and Tables

**Figure 1 microorganisms-11-00773-f001:**
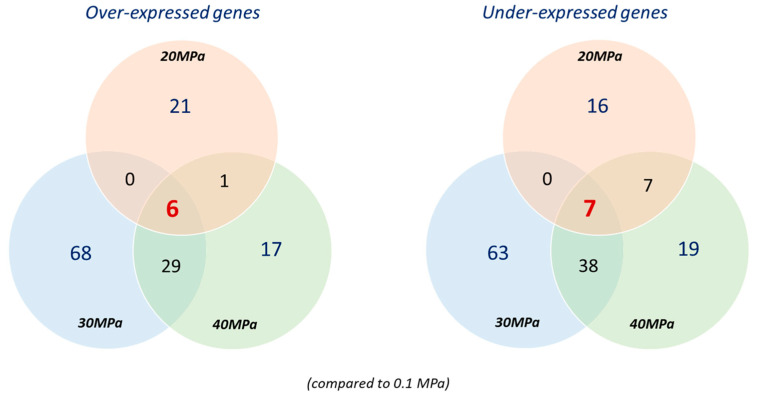
Venn diagram showing the number of significantly differentially expressed genes in the deep strain *P. elfii* DSM9442 when it was cultured at 20 MPa, 30 MPa, and 40 MPa compared to 0.1 MPa (adjusted *p*-value < 0.01, log_2_-fold change > 1.3).

**Figure 2 microorganisms-11-00773-f002:**
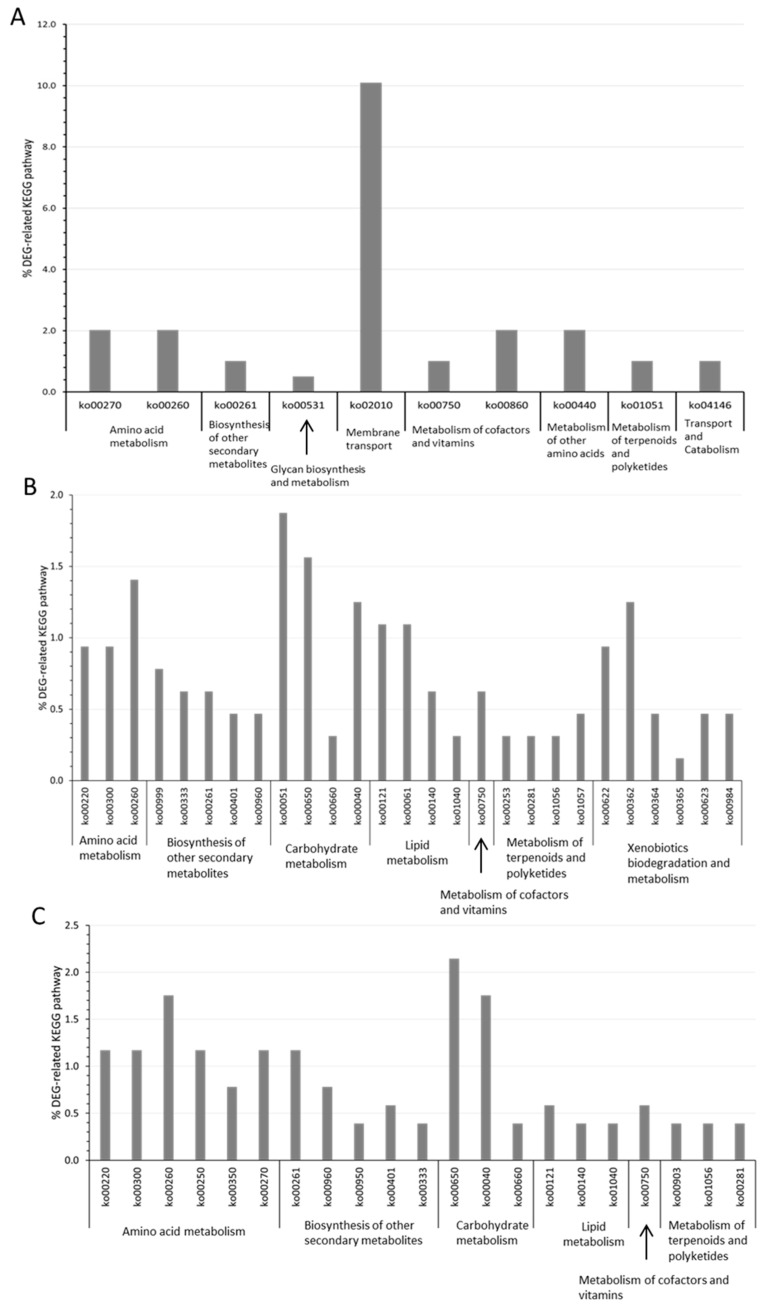
Significantly DEG-enriched KEGG pathways in the deep strain *P. elfii* DSM9442 cultured at (**A**) 20 MPa, (**B**) 30 MPa, and (**C**) 40 MPa compared to 0.1 MPa (one-tail z-test, *p*-value < 0.05). The *Y*-axis corresponds to the percentage of a specific KEGG pathway of all DEG-related KEGG pathways.

**Figure 3 microorganisms-11-00773-f003:**
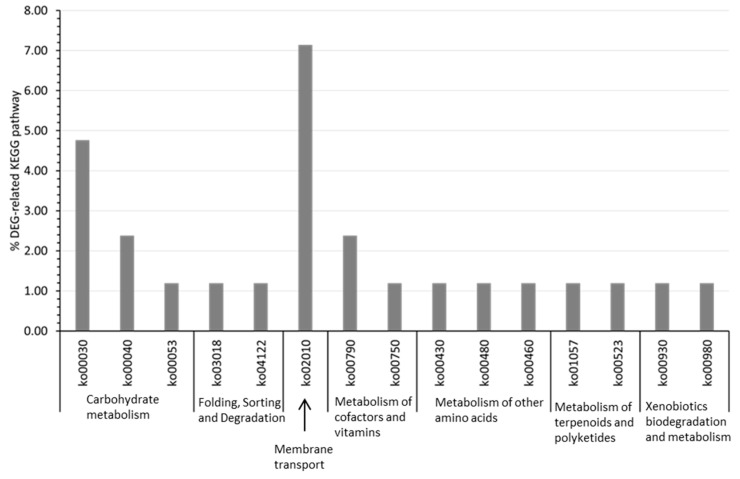
Significantly DEG-enriched KEGG pathways in the surface strain *P. elfii* subsp. *lettingae* cultured at 20 MPa compared to 0.1 MPa (one-tail z-test, *p*-value < 0.05). The *Y*-axis corresponds to the percentage of a specific KEGG pathway of all DEG-related KEGG pathways.

**Figure 4 microorganisms-11-00773-f004:**
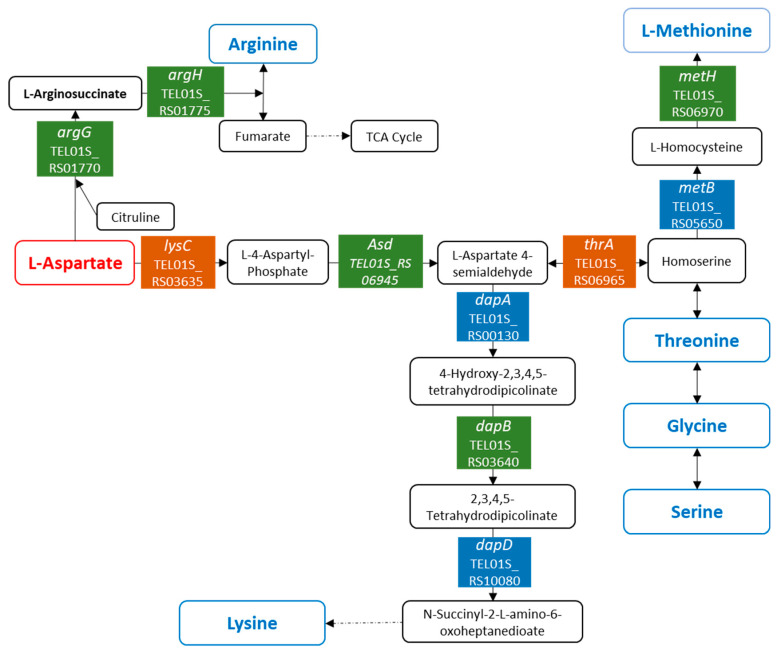
Schematic illustrating the amino acid metabolism pathways highlighted from the DEG. Colored boxes indicate the enzymes for which the genes are differentially expressed: blue means genes were over-expressed at 30 MPa, green means genes were over-expressed at 30 and 40 MPa, and orange means genes were over-expressed at 20, 30, and 40 MPa.

**Figure 5 microorganisms-11-00773-f005:**
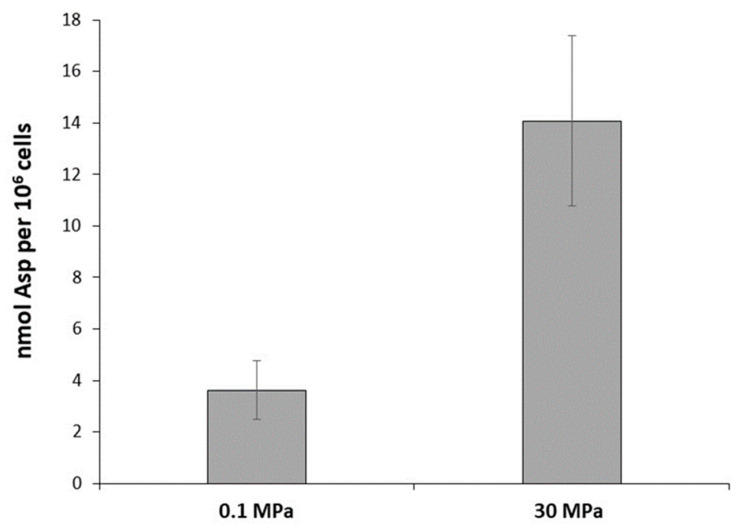
Intracellular aspartate concentration in *P. elfii* DSM9442 cultures grown at 0.1 MPa and 30 MPa. The values represent the means of assays from three independent cultures.

**Table 1 microorganisms-11-00773-t001:** List of the DEG involved in amino acid metabolism/transport in the deep strain *P. elfii* DSM9442.

Locus_Tag	KEGG Name	KEGG Pathway	DEG at 20 MPa *	DEG at 30 MPa *	DEG at 40 MPa *
TEL01S_RS01780	argC	ko00220:Arginine biosynthesis	0	−1	−1
TEL01S_RS01785	argJ	ko00220:Arginine biosynthesis	0	−1	−1
TEL01S_RS01770	argG	ko00220:Arginine biosynthesis|ko00250:Alanine, aspartate, and glutamate metabolism	0	−1	−1
TEL01S_RS01775	argH	ko00220:Arginine biosynthesis|ko00250:Alanine, aspartate, and glutamate metabolism	0	−1	−1
TEL01S_RS04500	glnA	ko00220:Arginine biosynthesis|ko00250:Alanine, aspartate, and glutamate metabolism	0	−1	−1
TEL01S_RS09130		ko00220:Arginine biosynthesis|ko00250:Alanine, aspartate, and glutamate metabolism|ko00270:Cysteine and methionine metabolism|ko00330:Arginine and proline metabolism|ko00350:Tyrosine metabolism|ko00360:Phenylalanine metabolism|ko00400:Phenylalanine, tyrosine, and tryptophan biosynthesis	0	1	1
TEL01S_RS03835	glmS	ko00250:Alanine, aspartate, and glutamate metabolism	0	−1	0
TEL01S_RS09135		ko00250:Alanine, aspartate, and glutamate metabolism|ko00260:Glycine, serine, and threonine metabolism	1	1	1
TEL01S_RS05590	soxA	ko00260:Glycine, serine, and threonine metabolism	0	−1	−1
TEL01S_RS10380	kbl	ko00260:Glycine, serine, and threonine metabolism	0	1	1
TEL01S_RS03635	lysC	ko00260:Glycine, serine, and threonine metabolism|ko00270:Cysteine and methionine metabolism|ko00300:Lysine biosynthesis	0	−1	−1
TEL01S_RS06945	asd	ko00260:Glycine, serine, and threonine metabolism|ko00270:Cysteine and methionine metabolism|ko00300:Lysine biosynthesis	0	−1	−1
TEL01S_RS06965	lysC_&_thrA	ko00260:Glycine, serine, and threonine metabolism|ko00270:Cysteine and methionine metabolism|ko00300:Lysine biosynthesis	−1	−1	−1
TEL01S_RS09745	trpB	ko00260:Glycine, serine, and threonine metabolism|ko00400:Phenylalanine, tyrosine, and tryptophan biosynthesis	0	−1	0
TEL01S_RS09140	serA	ko00260:Glycine, serine, and threonine metabolism	0	1	1
TEL01S_RS08685	mtnK	ko00270:Cysteine and methionine metabolism	−1	0	0
TEL01S_RS08690	mtnA	ko00270:Cysteine and methionine metabolism	−1	0	0
TEL01S_RS05650		ko00270:Cysteine and methionine metabolism|ko00260:Glycine, serine, and threonine metabolism	0	−1	0
TEL01S_RS06970	metH	ko00270:Cysteine and methionine metabolism	0	−1	−1
TEL01S_RS03640	dapB	ko00300:Lysine biosynthesis	0	−1	−1
TEL01S_RS00130	dapA	ko00300:Lysine biosynthesis	0	−1	0
TEL01S_RS09220	murF	ko00300:Lysine biosynthesis	0	1	0
TEL01S_RS10080	dapD	ko00300:Lysine biosynthesis|ko01230:Biosynthesis of amino acids	0	−1	0
TEL01S_RS08600	kdd	ko00310:Lysine degradation	0	1	0
TEL01S_RS05490	amiE	ko00330:Arginine and proline metabolism|ko00360:Phenylalanine metabolism|ko00380:Tryptophan metabolism	0	0	1
TEL01S_RS06380	hisC	ko00340:Histidine metabolism|ko00350:Tyrosine metabolism|ko00360:Phenylalanine metabolism|ko00400:Phenylalanine, tyrosine, and tryptophan biosynthesis	0	1	1
TEL01S_RS03685	iaaM	ko00380:Tryptophan metabolism	0	−1	0
TEL01S_RS05225	bglX	ko00460:Cyanoamino acid metabolism	0	1	1

* DEG with 0.1 MPa as reference: 0 means no significant gene variation, 1 means that expression at 0.1 MPa is significantly higher than at the indicated hydrostatic pressure, and −1 means that expression at 0.1 MPa is significantly lower than at the indicated hydrostatic pressure.

## Data Availability

The data presented in this study are openly available in the NCBI Gene Expression Omnibus (GEO, https://www.ncbi.nlm.nih.gov/geo/ (accessed on 25 February 2023)) under the accession number GSE226101.

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
