# Peer review of "Adaptation Strategies to High Hydrostatic Pressures in Pseudothermotoga species Revealed by Transcriptional Analyses"

_microorganisms, 2023, doi:10.3390/microorganisms11030773_

Round 1
Reviewer 1 Report
This work analyzed strain DSM9442 and strain DSM14385 response to different pressure. The topic is interesting, but English written needs to be further improved. Overall, minor revision is suggested.
1. Figure 1 used Venn diagram to show the number of genes that were under-expressed or over-expressed under pressures. The authors could consider adding a table or a supplementary file that lists the gene names, locus tags and their expression levels (under- or over-expressed) at each pressure condition. This would make it easier for readers to know the specific genes that are affected by pressure and their expression patterns.
2. The optimal growth pressure for the deep strain P. elfii DSM9442 is 20 MPa. It would make sense to use the gene expression or response of this strain at 20 MPa as a standard for comparison. The authors could provide more clarity on their rationale for using 0.1 MPa as the standard in their result section.
3. Can the authors provide a reason for why the data provided in Figure 2 and Figure 3 doesn’t have an error bar?
4. It is suggested to include a conclusion section, which summaries all results presented in the manuscript.
Author Response
We thank the reviewer for his(her) suggestions and comments. The revised version takes them into account and clarify the various points you mention. Below are the responses to your specific comments :
This work analyzed strain DSM9442 and strain DSM14385 response to different pressure. The topic is interesting, but English written needs to be further improved. Overall, minor revision is suggested.
The English written has been corrected by XpertScientific for both the first version and the revised version. The proofreading certificate is joined.
- Figure 1 used Venn diagram to show the number of genes that were under-expressed or over-expressed under pressures. The authors could consider adding a table or a supplementary file that lists the gene names, locus tags and their expression levels (under- or over-expressed) at each pressure condition. This would make it easier for readers to know the specific genes that are affected by pressure and their expression patterns.
Tables S3 and S4 contain all the genes with their locus-tag and annotations as well as their expression variations according to the culture pressure. It is possible to sort these tables according to different criteria and thus list the genes corresponding to the Venn diagram of Figure 1 easily. These tables allow any reader to search for any gene that was under or over expressed at a given pressure conditions. We thus do not think that adding another table will be informative, as all information is in Table S3 and S4
- The optimal growth pressure for the deep strain P. elfii DSM9442 is 20 MPa. It would make sense to use the gene expression or response of this strain at 20 MPa as a standard for comparison. The authors could provide more clarity on their rationale for using 0.1 MPa as the standard in their result section.
We agree with the reviewer that the optimal growth pressure for the P. elfii DSM9442 strain is 20 MPa. However, in order to make the comparison with the data obtained on P. elfii subsp. lettingae more understandable, we have chosen to keep the atmospheric pressure (0.1 MPa) as a reference and to analyze the gene expression changes observed at other pressures. It also seems to us that the understanding of the adaptation of cells to high hydrostatic pressures is easier if the reference is the atmospheric pressure.
- Can the authors provide a reason for why the data provided in Figure 2 and Figure 3 doesn’t have an error bar?
For this analysis, we wanted to know if a KEGG pathway was particularly affected by gene expression variations. To do this, we counted all KEGG pathways that involved genes whose expression varied at each hydrostatic pressure versus atmospheric pressure and compared this distribution to that of KEGG pathways identified in the genome of P. elfii DSM9445 and P. elfii subsp. lettingae. We then considered that a KEGG pathway was significantly enriched if its percentage was higher among the KEGG pathways linked to the DEGs than among those linked to the genomes. Therefore, it can't have an error bar on the bar plots, but the important fact is that the DEG-enriched KEGG pathways presented in the graphs of figures 2 and 3 are all statistically significant (z-test) with p-value <0.05.
- It is suggested to include a conclusion section, which summaries all results presented in the manuscript.
The manuscript includes the conclusion section lines 528-547 that summarizes the main results and messages of the article. We added a sentence to this paragraph about strain-specific and common traits of adaptation of the two strains (line 536-537).
We hope you will find this revised version of our manuscript suitable for publication now.
Sincerely Yours
Alain Dolla

Reviewer 2 Report
Please find the comments in the attached file.

Author Response
We thank the reviewer for his(her) suggestions and comments that are very helpful to improve our manuscript. Below are the responses to your specific comments:
- I’m not familiar with this genus, more background information about this genus in the introduction section would be helpful. For example, how many genomes are currently available in this genus for the moment? How is the environment where the two strains investigated in this study were isolated from?
Thanks for your suggestion. Additional information of the Pseudothermotoga genus as well as description of the biotopes where the studied strains have been isolated from, have been added line 58-72 in the revised version.
- The full name should be given when an abbreviation first appeared. For example, the “aspartate semialdehyde DH” in line 407, “Fas synthesis” in line 489, and I’m not sure if the word “toga” at line 501 is an abbreviation or not.
We agree with the reviewer. This has been checked throughout the text and corrected. The examples given by the reviewer have been corrected.
- Line 189-191 should be removed.
Done
- Please check the spelling of “poly-?-glutamate” in line 249.
Done. This was a typo. The real name is in poly-γ-glutamate and it has been corrected.
- Line 284 -285, please clarify if the expression of “specific transcripts” here refer to genes differentially expressed at one pressure but not other pressures, or gene specific identified in one strain but not the other.
We agree with the reviewer that “specific” is confusing. Actually, the comparison concerns the gene expression (quantity of transcripts) between 0.1 MPa and each of the other pressures. The term “specific” has been deleted and the sentence modified by "We conducted a systematic comparison between detected transcripts of P. elfii DSM9442 cultured at 0.1 MPa (atmospheric pressure) versus detected transcripts when grown at 20 MPa (optimal pressure for growth), 30 MPa, and 40 MPa (maximum pressure for growth)”.
- It is interesting that genes involved in amino acid metabolism consuming aspartate, as well as genes involved in aspartate import were up-regulated at 30 MPa (as shown in figure 4), the intracellular concentration of aspartate increased. Does it suggest that the transport of Asp is more efficient than its consumption? It could be further discussed/explained.
From our transcriptomic data, we have no evidence of an increase of the aspartate biosynthesis at high hydrostatic pressure. As the reviewer says, accumulation of aspartate at high hydrostatic pressure suggests that its import is greater than its consumption. It is to be noted that in the experiments described here, the sources of carbon and energy are peptides and/or amino acids; thus , the transport systems need to be highly efficient to provide the carbon and energy required for cell growth. This point has been added in the revised manuscript lines 467-470.
- More interpretations and explanations are required to support the linkage between up-regulated amino acid metabolism to the chain formation of P.elfii cells.
Our transcriptomic data suggest that at high hydrostatic pressure, the feeding of the urea cycle from aspartate if boosted; this leads to the synthesis of arginine and of fumarate also, which, in turn, could contribute to gluconeogenesis via the production of oxaloacetate through the TCA cycle. It has been already reported that aspartate is a glucogenic amino acid. The end-product of the gluconeogenesis can be used to synthetize molecules required for the cell envelope. In P. elfii strains, we reported that the increase of the hydrostatic pressure led to the formation of chains composed of up to 15 cells per chains and that the chain formation should be considered as a protective mechanism rather than a degenerative phenomenon (Roumagnac et al, 2020). We thus propose that the boost at high hydrostatic pressure of gluconeogenesis from aspartate would allow the synthesis of the molecules required for cell envelope, assisting the chain formation. This point has been clarified in the revised version lines 439-452.
We hope you will find the revised version of our manuscript now suitable for publication.
Sincerely Yours
Alain Dolla